# Effect of Combined Levothyroxine (L-T_4_) and 3-Iodothyronamine (T_1_AM) Supplementation on Memory and Adult Hippocampal Neurogenesis in a Mouse Model of Hypothyroidism

**DOI:** 10.3390/ijms241813845

**Published:** 2023-09-08

**Authors:** Grazia Rutigliano, Andrea Bertolini, Nicoletta Grittani, Sabina Frascarelli, Vittoria Carnicelli, Chiara Ippolito, Stefania Moscato, Letizia Mattii, Claudia Kusmic, Alessandro Saba, Nicola Origlia, Riccardo Zucchi

**Affiliations:** 1Institute of Clinical Science, Imperial College London, London SW7 2AZ, UK; 2Department of Pathology, University of Pisa, 56126 Pisa, Italy; a.bertolini2@student.unisi.it (A.B.); nicolettamaria.grittani@gmail.com (N.G.); sabina.frascarelli@unipi.it (S.F.); vittoria.carnicelli@unipi.it (V.C.); alessandro.saba@unipi.it (A.S.); riccardo.zucchi@unipi.it (R.Z.); 3CNR Institute of Clinical Physiology, 56124 Pisa, Italy; kusmic@ifc.cnr.it; 4Department of Clinical and Experimental Medicine, University of Pisa, 56126 Pisa, Italy; chiara.ippolito@unipi.it (C.I.); stefania.moscato@unipi.it (S.M.); letizia.mattii@unipi.it (L.M.); 5CNR Institute of Neuroscience, 56124 Pisa, Italy; nicola.origlia@in.cnr.it

**Keywords:** methimazole, novel object recognition, thyroperoxidase, mass spectrometry, doublecortin, neuroendocrinology

## Abstract

Mood alterations, anxiety, and cognitive impairments associated with adult-onset hypothyroidism often persist despite replacement treatment. In rodent models of hypothyroidism, replacement does not bring 3-iodothyronamine (T_1_AM) brain levels back to normal. T_1_AM is a thyroid hormone derivative with cognitive effects. Using a pharmacological hypothyroid mouse model, we investigated whether augmenting levothyroxine (L-T_4_) with T_1_AM improves behavioural correlates of depression, anxiety, and memory and has an effect on hippocampal neurogenesis. Hypothyroid mice showed impaired performance in the novel object recognition test as compared to euthyroid mice (discrimination index (DI): 0.02 ± 0.09 vs. 0.29 ± 0.06; t = 2.515, *p* = 0.02). L-T_4_ and L-T_4_+T_1_AM rescued memory (DI: 0.27 ± 0.08 and 0.34 ± 0.08, respectively), while T_1_AM had no effect (DI: −0.01 ± 0.10). Hypothyroidism reduced the number of neuroprogenitors in hippocampal neurogenic niches by 20%. L-T_4_ rescued the number of neuroprogenitors (mean diff = 106.9 ± 21.40, t = 4.99, p_corr_ = 0.003), while L-T_4_+T_1_AM produced a 30.61% rebound relative to euthyroid state (mean diff = 141.6 ± 31.91, t = 4.44, p_corr_ = 0.004). We performed qPCR analysis of 88 genes involved in neurotrophic signalling pathways and found an effect of treatment on the expression of *Ngf*, *Kdr*, *Kit*, *L1cam*, *Ntf3*, *Mapk3*, and *Neurog2*. Our data confirm that L-T_4_ is necessary and sufficient for recovering memory and hippocampal neurogenesis deficits associated with hypothyroidism, while we found no evidence to support the role of non-canonical TH signalling.

## 1. Introduction

Thyroid hormones (TH) play an essential role in brain development during gestational and perinatal periods and contribute to maintaining brain health in adulthood [1]. Adult-onset hypothyroidism is associated with depression, anxiety [2], and memory deficits [3,4] both in humans and in animal models [5,6,7,8,9]. Poor memory performance is accompanied by altered synaptic plasticity in hippocampal circuits [10,11,12] and impaired neurogenesis in the neurogenic niche of the sub-granular zone (SGZ) of the dentate gyrus [13,14,15,16]. Adult neurogenesis is highly sensitive to the hormonal, neurotransmitter, and growth factor milieu within the neurogenic niche, including TH signalling [7,12,17]. Levothyroxine (L-T_4_) monotherapy is the treatment of choice for hypothyroidism [18]. However, about 10–15% of patients complain of persisting symptoms of TH deficiency, such as cognitive disturbances, fatigue, mood swings, and anxiety [19], despite thyroid-stimulating hormone (TSH) levels being within the normal range. Serum TSH may not reflect intracellular TH levels [19,20]. Additionally, generation of triiodothyronine (T_3_) from thyroxine (T_4_) by deiodinases may not be equivalent to thyroidal secretion of T_3_ [21]. Finally, we hypothesise that persistent symptom of hypothyroidism may be due to decreased brain levels of downstream TH derivatives, in particular, 3-iodothyronamine (T_1_AM). In fact, in mice treated with methimazole—a thyroperoxidase inhibitor—and L-T_4_-replacement, brain T_1_AM levels remained undetectable despite an almost full recovery of free T_4_ and T_3_ levels (fT_4_ and fT_3_) [22]. T_1_AM is an endogenous trace amine involved in the regulation of metabolic homeostasis and central nervous system (CNS) function [23]. T_1_AM showed pro-cognitive and anti-amnestic effects [24], and neuroprotective actions in several models, including seizure-related excitotoxic damage, altered autophagy, amyloidosis, and ischemia–reperfusion injury [25,26,27,28]. It is unknown if T_1_AM participates in the regulation of adult hippocampal neurogenesis. Here, we investigated whether augmenting L-T_4_ treatment with T_1_AM improves neurocognitive and neurobiological alterations associated with adult-onset hypothyroidism and has an effect on hippocampal neurogenesis. In a rodent pharmacological model of hypothyroidism, we compared performance in memory, locomotion, and anxiety- and depression-related tasks in response to different replacement strategies combining L-T_4_ and T_1_AM. We evaluated the effect of these replacement strategies on neurogenetic capacity in the SGZ using histological markers and the expression of genes involved in neurogenetic pathways.

## 2. Results

### 2.1. Generation of Rodent Model of Hypothyroidism

The experimental protocol is depicted in Figure 1A. Control euthyroid mice (*n* = 17) had an initial average body weight (BW) of 21.38 ± 0.45 g at day 0, which increased over the 3 weeks to 25.88 ± 0.46 g. BW changed less in hypothyroid mice (*n* = 15), from 22.48 ± 0.18 g to 23.89 ± 0.18 g. Two-way ANOVA analysis found a statistically significant interaction between treatment and time on BW (F(3, 279) = 35.21, *p* < 0.0001). Hypothyroid mice treated with L-T_4_ (*n* = 18, final BW: 26.46 ± 0.42 g) and L-T_4_+T_1_AM (*n* = 17, final BW: 26.79 ± 0.45 g) partially recovered BW, while BW of T_1_AM-treated (*n* = 15, final BW: 25.58 ± 0.45 g) and untreated hypothyroid mice (*n* = 15, final BW: 26.0 ± 0.49 g) remained significantly lower than that of euthyroid mice (*n* = 17, final BW: 27.79 ± 0.46 g). The two-way ANOVA showed a significant effect of replacement treatment (F(4, 542) = 11.66, *p* < 0.0001), time (F(7, 542) = 54.37, *p* < 0.0001), and their interaction (F(28, 542) = 1.63, *p* = 0.022) on BW (Figure 1B).

### 2.2. Serum Levels of TT_4_ but Not TT_3_ Were Different between the Experimental Groups

Hypothyroidism almost completely suppressed total T_4_ levels(TT_4_) as compared to euthyroid mice (2.53 ± 0.12 ng/mL and 32.77 ± 1.48 ng/mL, *p* < 0.01). Serum TT_4_ levels were restored by L-T_4_ and L-T_4_+T_1_AM (38.92 ± 10.49 ng/mL and 38.88 ± 6.13 ng/mL) but not by T_1_AM (3.40 ± 1.01 ng/mL) (Figure 1C). We observed no significant differences in total T_3_ levels (TT_3_)between treatments (Figure 1D).

### 2.3. Behavioural Correlates of Hypothyroidism and Replacement Treatments

Data relative to behavioural measures are reported in Table 1. In the elevated plus maze (EPM), hypothyroid mice showed a significantly lower number of stretch–attend postures (SAP), indicating reduced risk-assessment behaviours compared to euthyroid mice (t = 3.03, p_corr_ = 0.006, surviving Bonferroni correction for multiple comparison). None of the other EPM measures were significantly different between hypothyroid and euthyroid mice. One-way ANOVA did not detect any significant differences in primary and secondary anxiety-related behaviours or decision-making and risk-assessment measures between treatments (Appendix A). 

We did not find any significant differences in spontaneous locomotor activity and thigmotaxis in the open field test (OF) or percentage immobility time in the tail suspension test (TST) between treatments (Figure 2 and Appendix A). Hypothyroid mice showed significantly lower discrimination index (DI) in the ORT (DI: hypothyroid: 0.02 ± 0.09; euthyroid: 0.29 ± 0.06; t = 2.515, *p* = 0.02, Figure 3). One-way ANOVA revealed a significant effect between treatments (F(3, 42) = 3.15, *p* = 0.03). Subsequent post-hoc tests only detected a significant difference between hypothyroid and L-T_4_+T_1_AM-treated mice (mean diff. = −0.31, *p* < 0.05). The visual inspection of the DI does, however, suggest an almost complete recovery of memory in L-T_4_-treated relative to hypothyroid mice, while T_1_AM had no effect (DI: −0.01 ± 0.10). DIs diverged significantly from a hypothetical value of 0 (no discrimination) in euthyroid (t = 4.82, *p* < 0.001), L-T_4_-treated (t = 3.53, *p* < 0.01), and L-T_4_+T_1_AM-treated (t = 4.04, *p* < 0.01) mice, confirming the ability to discriminate the novel object and therefore intact memory in these groups (Figure 3). 

### 2.4. L-T_4_ and L-T_4_+T_1_AM Restored the Number of DCX+ Cells 

Hypothyroidism induced a 20% reduction in the number of doublecortin-positive (DCX+) cells (mean diff = −53.50 ± 23.81, t = 2.25, p_uncorr_ = 0.07). One-way ANOVA revealed a trend toward a significant effect of treatment (F(4, 15) = 2.79, *p* = 0.06). We performed pair-wise comparisons between the five groups using a *t*-test, using Bonferroni correction for multiple comparisons (0.05/10 = 0.005). Compared to hypothyroidism, L-T_4_ increased the number of cells by 45.58% (mean diff = 106.9 ± 21.40, t = 4.99, p_corr_ = 0.003) while L-T_4_+T_1_AM produced a 60.44% increase (mean diff = 141.6 ± 31.91, t = 4.44, p_corr_ = 0.004), the latter corresponding to a 30.61% rebound relative to euthyroidism (mean diff = 88.13 ± 36.99, t = 2.38, p_uncorr_ = 0.05). T_1_AM had no effect (mean diff = 72.13 ± 58.84, t = 1.23, p_uncorr_ = 0.27) (Figure 4). These results confirm that L-T_4_ is a crucial endocrine signal controlling progenitor development within neurogenic niches, while there are no significant effects of T_1_AM in the modulation of adult hippocampal neurogenesis. 

### 2.5. L-T_4_+T_1_AM Enhanced the Expression of Genes Involved in Neurogenetic Pathways

The qPCR analysis revealed differentially expressed genes in neurogenetic pathways. One-way ANOVA revealed a significant effect of treatment on expression of *Kdr* (F(4, 24) = 3.20, *p* = 0.03), *L1cam* (F(4, 24) = 2.84, *p* = 0.04), *Mapk3* (F(4, 23) = 2.96, *p* = 0.04) and *Neurog2* (F(4, 20) = 3.94, *p* = 0.02). We observed a trend towards an effect of treatment on expression of *Ngf* (F(4, 21) = 2.50, *p* = 0.07), *Kit* (F(4, 24) = 2.51, *p* = 0.07) and *Ntf3* (F(4, 23) = 2.51, *p* = 0.07). We performed *t*-test pair-wise comparisons to test the differences between L-T_4_+T_1_AM and either L-T_4_ or T_1_AM, using Bonferroni correction for multiple comparisons (0.05/14 = 0.004). We found that *Mapk3* and *Neurog2* were significantly upregulated in L-T_4_+T_1_AM-treated relative to L-T_4_-treated mice (*Mapk3*, t = 4.51, p_corr_ = 0.002; *Neurog2*, t = 4.81, p_corr_ = 0.001), with non-significant differences between L-T_4_+T_1_AM and T_1_AM (*Mapk3*, t = 0.16, p_uncorr_ = 0.87; *Neurog2*, t = 1.13, p_uncorr_ = 0.29), suggesting that changes in gene expression could be induced by T_1_AM on its own (Figure 5). 

## 3. Discussion

In this study, we confirmed that L-T_4_ monotherapy is necessary and sufficient to recover deficits in memory and adult hippocampal neurogenesis associated with adult-onset hypothyroidism. We observed no further improvement in neurocognitive and neurobiological alterations from augmenting L-T_4_ with T_1_AM.

In our model of adult-onset hypothyroidism, we found no alterations in behavioural measures of locomotion, anxiety, or depression, in contrast with empirical clinical observation and previous reports [7,29,30,31]. Our hypothyroidism induction protocol almost completely suppressed serum TT_4_ levels, while we observed no changes in serum TT_3_ levels, suggesting that our protocol induced moderate hypothyroidism. Residual thyroperoxidase activity may be able to ensure a sufficient production of T_3_ to prevent alteration in measures of depression or anxiety.

On the other hand, our model of moderate hypothyroidism showed alterations in memory, suggesting that cognitive functions are more susceptible to TH deficiency. The effect of TH deficit on cognitive functions and synaptic plasticity has been extensively demonstrated in rodents [7,8,9,10,32] and corresponds to clinical evidence of deficits in spatial, associative and verbal memory in hypothyroid patients [3,4]. In our hypothyroid mice, we found a 20% decrease in newborn neuroblasts in the SGZ of the dentate gyrus. A 4-week treatment with L-T_4_ was necessary and sufficient to recover both memory and number of neuroblasts, consistent with previous evidence of the rescuing effects of L-T_4_ on hypothyroidism-induced behavioural, electrophysiological, morphological, and molecular alterations [7,9,17,32]. TH play a crucial role toward the postmitotic survival of adult dentate granule cell progenitors [17]. It remains unclear if the decline in survival is due to a delay/block in neuronal differentiation or whether these cells may be targeted for cell death. The susceptibility of cognitive functions and neurogenesis to moderate TH changes may explain the relationship between TH and neurodegenerative diseases, such as Alzheimer’s disease, observed in clinical populations [33,34,35], including those with subclinical hypothyroidism [2].

We found no significant differences between L-T_4_ monotherapy and L-T_4_+T_1_AM in memory and number of DCX+ cells. T_1_AM did not show any effect per se. We focused on T_1_AM because we hypothesise that some of the neurocognitive symptoms of hypothyroidism traditionally attributed to lack of TH are partly caused by reduced availability of downstream metabolites. It has been demonstrated that T_1_AM remains undetectable in the tissues of animals pharmacologically depleted of TH, even following full recovery of fT_4_ and fT_3_ [22]. Additionally, evidence has been accumulating about the effects of T_1_AM in the CNS. T_1_AM is a modulator of noradrenergic, dopaminergic, and histaminergic transmission [23]. It has been observed to modulate sleep [36], exploratory activity, memory acquisition and retention [24]. T_1_AM showed neuroprotective actions in models of seizure-related excitotoxic damage, altered autophagy, amyloidosis, ischemia–reperfusion injury, and neuroinflammation [25,26,27,28,37]. In these studies, T_1_AM was administered either through perfusion on ex vivo brain acute/organotypic slices or through intracerebral microinjection, while we used a systemic administration. Studies employing radioactively labelled T_1_AM showed that T_1_AM crosses the blood–brain barrier [38].

We found an effect of replacement treatment on expression of several genes, including *Ngf*, *Kdr*, *Kit*, *L1cam*, *Ntf3*, *Mapk3*, and *Neurog2*. In particular, L-T_4_+T_1_AM-treated mice displayed an upregulation of *Mapk3* and *Neurog2* as compared to L-T_4_-treated mice. However, we did not observe significant differences between L-T_4_+T_1_AM and T_1_AM in *Mapk3* and *Neurog2* expression, suggesting that changes in gene expression could be induced by T_1_AM on its own. These genes are involved in neurogenetic mechanisms, such as increased proliferation of DCX+ cells in the hippocampus [39], initiation, and progression through the G1 phase of the cell cycle [40]; neuronal cell faith determination [40]; and suppression of glial faith promoting neurogenesis [41]. TH regulate gene transcription through thyroid receptors (TR), widely expressed in different stages of granule cells progenitors [17]. T_1_AM does not bind to TR but to several molecular targets, including the trace amine-associated receptor 1 (TAAR1), alpha2 adrenergic receptors, transient receptor potential channels, and ApoB-100 [42]. How its distinct target profile links to gene expression will need to be clarified in future studies. However, we hypothesise a role for TAAR1 due to growing evidence of its involvement in neuropsychiatric disorders [43,44]. Furthermore, we have previously demonstrated that T1AM neuroprotective actions in models of amyloidosis, ischemia–reperfusion injury, and neuroinflammation are dependent on TAAR1 [27,28,37]. TAAR1 was reported to activate distinct signalling pathways depending on its localization in different cellular compartments. When on the plasma membrane, TAAR1 activation leads to increased intracellular cAMP levels and activation of the protein kinase A through coupling to Gαs proteins and adenylyl cyclase activation [45,46,47]. Intracellular TAAR1 activation induces its coupling with Gα13 subunits to activate the GTPase RhoA [48]. TAAR1 activation is also associated with G protein-coupled inwardly rectifying potassium channels [49,50] and G-protein-independent inhibition of glycogen synthase kinase 3β [51].

The present study has some limitations. First, our findings do not clarify if the effects of replacement treatments on memory are causally mediated by changes in hippocampal neurogenesis. Future studies should investigate if blocking neurogenesis during L-T4 and/or L-T_4_+T_1_AM treatment prevents memory rescue. Second, for some comparisons, the sample size was not sufficient to have enough statistical power. This is probably due to our protocol inducing moderate hypothyroidism (compensated TT_3_ levels), which may have caused behavioural changes of small-to-moderate effect size, while our statistical analysis was powered to detect larger effect sizes, based on previous reports in thyroidectomized or transgenic rodents. Finally, to test our hypothesis that some of hypothyroidism-induced cognitive impairments are explained by T_1_AM deficiency, we planned to measure brain T_1_AM in different treatments. However, similarly to others [52], we found no clear endogenous signal of T_1_AM (below detection limit) in most of our pilot samples. Due to these technical reasons, we therefore decided to abort this objective.

Our data show that moderate TH changes are sufficient to induce significant impairments in memory and neurogenesis. L-T_4_ was necessary and sufficient to recover both memory and number of neuroprogenitors to the euthyroid state, while we found no evidence to support the role of non-canonical TH signalling. This was despite an effect of T_1_AM on expression of genes involved in neurogenetic pathways, which should be further explored in future studies.

## 4. Materials and Methods

### 4.1. Mouse Model of Pharmacological Hypothyroidism

Six-week-old C57BL/6J male mice (maintained in a 12 h hight/dark cycle with ad libitum access to diet and water) were given methimazole (0.20 mg/g/day) and potassium perchlorate (0.30 mg/g/day) [22] in drinking water for 49 days, while control littermates received water. At day 21, mice were anesthetised with 1.5% isoflurane and implanted with ALZET^®^ subcutaneous osmotic pumps (Charles River Laboratories, Milan, Italy) delivering replacement treatments for 28 days (reservoir volume of 100 μL, delivery rate of 0.11 μL/h). Mice were divided into 5 groups: euthyroid, hypothyroid, L-T_4_ (0.04 μg L-T_4_/g BW/day [53]), L-T_4_+T_1_AM (0.04 μg L-T_4_ + 0.004 μg T_1_AM/g BW/day) and T_1_AM (0.004 μg T_1_AM/g BW/day) [27] (Figure 1A). All drugs were purchased from Sigma Aldrich-Merck (Milan, Italy). BW was monitored weekly throughout treatment. All experiments were conducted in accordance with the guidelines for the care and use of laboratory animals of the Italian Ministry of Health (Legislative Decree n. 116/92) and the European Community (European Directive 86/609/EEC). The Italian Ministry of Health approved the use of animals in this protocol (65E5B.10, n.734/2017-PR, 10 October 2017).

### 4.2. HPLC-MS/MS Analysis of Thyroid Hormones

Serum samples were processed to perform a HPLC-MS/MS quantification of TT_3_ and TT_4_ based on the previously reported methods [54] described in the Appendix A.

### 4.3. Behavioural Tests

Before testing, mice were habituated to experimental handling daily for 10 min. Over the last 3 days before the experiment, mice were habituated to the testing room for 1 h, i.e., the experimenter moved them from their usual holding room to the testing room, a soundproof environment with 100 lux illumination level. Afterwards, mice underwent behavioural tests over three consecutive days, in the following order: day 1, EPM; day 2, OF and ORT; day 3, TST. Mouse behaviour was videotaped and scored manually by investigators blind to the experimental condition. For the automatic scoring of the OF, we used the toolbox developed by Patel [55].

#### 4.3.1. Elevated Plus Maze Test

The EPM apparatus consists of 4 arms (25 cm length × 5 cm width) forming a plus sign, elevated 50 cm above the floor. Two opposite arms have walls (16 cm height, closed arms), while the other two do not (open arms) (Appendix A). Mice were placed in the centre and allowed to freely explore the maze for 5 min. Recorded variables and their interpretation [56,57,58,59] are detailed in Table 2.

#### 4.3.2. Open Field Test

Mice were allowed to freely explore the OF arena for 10 min (Figure 2A). We scored total travelled distance, and thigmotaxis, i.e., time spent in outer zones versus inner zones.

#### 4.3.3. Novel Object Recognition Test

After a 10 min habituation, mice were allowed to freely explore two identical objects for 3 min (familiarization). Then, one of the two objects was replaced with a novel object. In the test phase, mice were left to freely explore the objects for 3 min (Figure 3A). Exploration time was counted when the mice were within 2 cm from the object with the nose directed towards it, sniffing, or touching the object with the nose but not when the nose was pointing away from the object even if the mice were beside the object, running around it, sitting, or climbing on it. DI were calculated according to the formula:DI = (time at novel − time at familiar)/(time at novel + time at familiar)(1)

#### 4.3.4. Tail Suspension Test

Mice were suspended from a shelf, with climb-stoppers around the tail (Appendix A). Escape-related behaviours were counted when mice: (a) tried to reach the apparatus walls and suspension bar; (b) shook the body; (c) showed movements akin to running. Escape was not counted when mice showed small movements confined to the front legs without involvement of the hind legs or in cases of oscillations due to the momentum gained during mobility bouts. Lack of escape-related behaviour was considered as immobility and was taken as a measure of depression.

### 4.4. Immunofluorescence

We performed immunofluorescence for Ki67 for cell proliferation and DCX for newborn Type 2b and Type 3 neuroblasts. After transcardial perfusion with 4% paraformaldehyde, brains were removed, cryoconserved in 30% sucrose and stored at −80 °C. We cut 10 μm thick coronal sections using a cryostat (Leica CM3050 S, Leica Biosystems, Milan, Italy) at 180 μm intervals across the hippocampus (−1.82 mm to −2.36 mm from bregma). For antigen retrieval, we immersed the slices in sodium citrate pH 6.0 at 85 °C three times for 1 min each time. Slices were hydrated in phosphate buffer saline (PBS, Sigma Aldrich-Merck) for 20 min at room temperature (RT) and washed twice with 0.3% Triton™ X-100 (Sigma Aldrich-Merck) in PBS. Following a one-hour blocking step in 5% bovine serum albumin (BSA) and 0.5% Triton™ X-100 solution in PBS at RT, hippocampal slices were incubated with anti-Ki67 (Rat monoclonal, #14569880, Life Technologies, Monza, Italy) (1:500) and anti-DCX (Rabbit, #Ab18723, Abcam, Cambridge, UK) (1:800) primary antibodies (dissolved in 0.1% BSA, 0.1% Triton™ X-100 in PBS, Sigma Aldrich-Merck) at 4 °C overnight. After washing, slides were incubated for 2 h at RT, in the dark, with goat anti-rat Alexa-488-conjugated (1:1000, Invitrogen, Waltham, MA, USA, #A11006) and donkey anti-rabbit Alexa-647-conjugated (1:1000, Abcam, #Ab150075) secondary antibodies (dissolved in 1% BSA and 0.1% Triton™ X-100 in PBS). Slices were mounted with the Fluoroshield™ medium containing DAPI (Abcam, #Ab104139) for staining nuclei. Images were captured using a Leica TCS SP8 Laser Scanning Confocal Microscope (Leica Biosystems) and analysed using Fiji 2.9.0.

### 4.5. Gene Expression Analysis

RNA was isolated from hippocampi dissected and homogenized in TRIzol™ (ThermoFisher Scientific, Milan, Italy). RNA yield was determined with the Qubit Fluorometer (ThermoFisher Scientific). After DNase digestion, RNA was retrotranscribed using the iScript Clear cDNA Synthesis kit (Bio-Rad, Coralville, IA, USA). qPCR analysis was performed in PrimePCR^TM^ “Neurogenesis Tier 1 M96” collection panel (Bio-Rad), a predesigned 96-well PCR plate containing primer sets for 88 gene targets involved in neurogenesis pathway for use with SYBR^®^ Green (Appendix A) on a CFX Connect Real Time System (BioRad). Ct values > 35 were considered to be no expression. Fold change calculation by ΔΔCt method [60] was performed with the CFX Maestro™ Software version 2.2 (Bio-Rad, Hercules, CA, USA) using three reference genes for normalization: TATA-binding protein (*Tbp*), glyceraldehyde-3-phosphate dehydrogenase (*Gadph*), and Hypoxanthine-guanine phosphoribosyltransferase (*Hprt1*).

### 4.6. Statistical Analysis

Outliers were removed with the ROUT method (Q = 1%). Normal distribution was assessed with the D’Agostino and Pearson test. Data were reported as mean ± standard error of the mean (SEM) or as median and interquartile ranges as appropriate. BW were analysed using two-way ANOVA for repeated measures, followed by Sidak’s post hoc tests. Differences between groups in hormonal level, behavioural performance, immunofluorescence, and gene expression were assessed with one-way ANOVA, followed by Tukey’s post-hoc tests or with a Kruskal–Wallis test as appropriate. For the ORT, one-sample *t*-test was used to determine whether the average DI for each group was different from chance (hypothetical value = 0). Statistical significance was set at *p* < 0.05. Statistical analyses were conducted using GraphPad Prism 6 (GraphPad Software Inc., La Jolla, CA, USA).

## Figures and Tables

**Figure 1 ijms-24-13845-f001:**
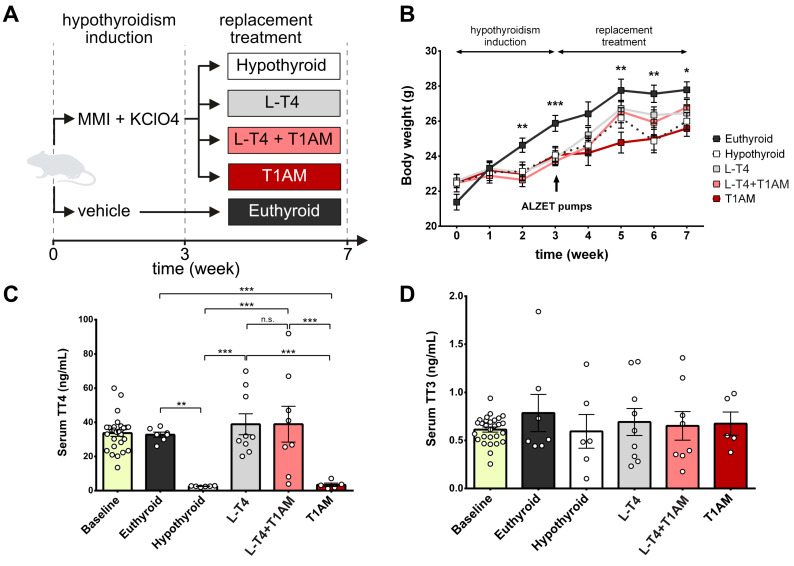
Murine model of pharmacological hypothyroidism. (**A**) Experimental protocol. Animals were divided into 5 groups: euthyroid (*n* = 17), hypothyroid (*n* = 15), hypothyroid treated with L-T_4_ (*n* = 18), hypothyroid treated with L-T_4_+T_1_AM (*n* = 17), and hypothyroid treated with T_1_AM (*n* = 15). MMI, methimazole; KClO4, potassium perchlorate. (**B**) Body weight (BW) changes throughout the induction of hypothyroidism and replacement treatment. BW of hypothyroid mice started to diverge from euthyroid mice at week 2. Hypothyroid mice treated with L-T_4_ and L-T_4_+T_1_AM partially recovered BW, while T_1_AM-treated and untreated hypothyroid mice remained significantly lighter than euthyroid mice. Two-way ANOVA testing the effect of treatment, time and their interaction and Tukey’s post-hoc comparison were used to test differences between the groups (* *p* < 0.05, ** *p* < 0.01, *** *p* < 0.001). (**C**) Serum TT_4_ levels were almost completely suppressed in the hypothyroid group and restored by L-T_4_ and L-T_4_+T_1_AM. (**D**) We observed no significant changes in serum TT_3_ levels between the groups. ANOVA and Tukey’s multiple comparison were used to test differences between the groups (** *p* < 0.01, *** *p* < 0.001, n.s. not significant). Results are plotted as mean ± SEM using GraphPad Prism 6 software. Baseline *n* = 28, hypothyroid *n* = 6, euthyroid *n* = 7, L-T_4_
*n* = 9, L-T_4_+T_1_AM *n* = 8, T_1_AM *n* = 6. The remaining samples were used to analyse serum fT_4_ and fT_3_ on an AIA 2000LA immunoassay platform.

**Figure 2 ijms-24-13845-f002:**
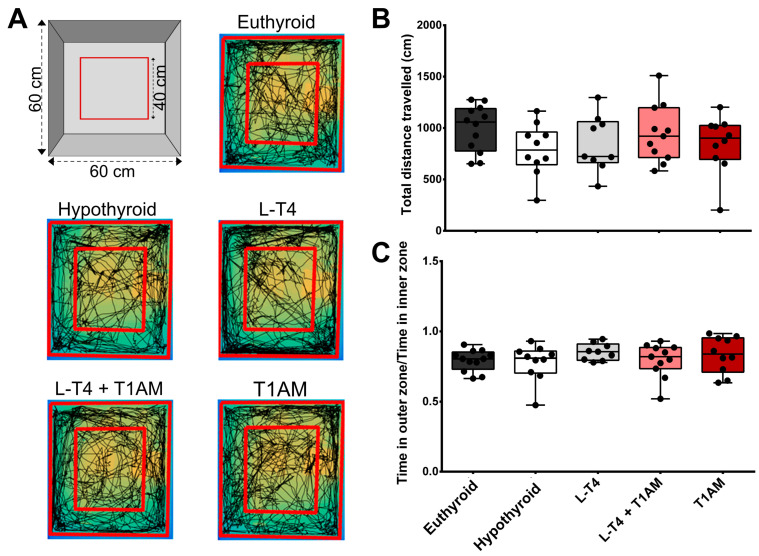
Effect of hypothyroidism and different replacement treatments on locomotion. (**A**) Open field apparatus and example trajectories in the open field test (10 min duration) for experimental mice in the five experimental groups. We found no between-group differences in total distance travelled (**B**) or thigmotaxis (**C**) in the open field test. Hypothyroid *n* = 10, euthyroid *n* = 12, L-T_4_
*n* = 9, L-T_4_+T_1_AM *n* = 11, T_1_AM *n* = 10. Results are plotted as mean ± SEM using GraphPad Prism 6 software.

**Figure 3 ijms-24-13845-f003:**
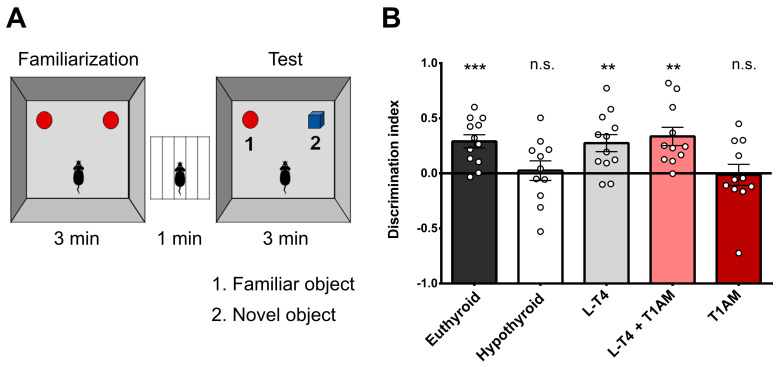
Effect of hypothyroidism and different replacement treatments on memory. (**A**) Graphical representation of the apparatus used for the novel object recognition test (ORT). (**B**) One sample *t*-test comparing DIs of the five experimental groups to a hypothetical value of 0 (no discrimination of the novel vs. familiar object) showed that DIs diverged significantly from 0 in euthyroid, L-T_4_-treated and L-T_4_+T_1_AM-treated mice, confirming intact memory in these groups (** *p* < 0.01, *** *p* < 0.001, n.s. not significant). Results are plotted as mean ± SEM using GraphPad Prism 6 software. Hypothyroid *n* = 11, euthyroid *n* = 12, L-T_4_
*n* = 12, L-T_4_+T_1_AM *n* = 11, T_1_AM *n* = 11.

**Figure 4 ijms-24-13845-f004:**
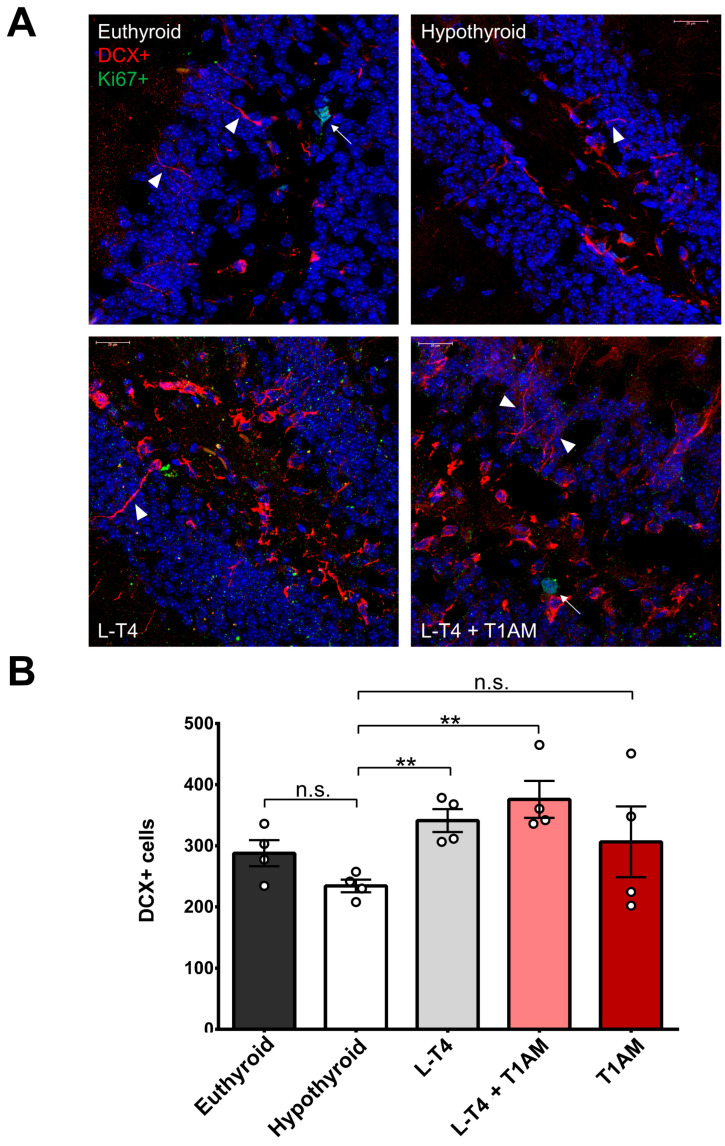
The effect of different replacement treatments on hippocampal neurogenesis. (**A**) Representative confocal microscopy images of sections illustrating immunofluorescence of DCX+ (red) and Ki67+ (green). DAPI was used to counterstain nuclei (blue). The arrows point at proliferating cells, and the arrowheads point at some neuroblast neurites. Scale bars 20 µm. (**B**) DCX+ cell count. Each point represents the sum of DCX+ cells from four coronal sections taken across the hippocampus of a single mouse. Hypothyroidism caused a 20% reduction in the number of DCX+ cells in hippocampal neurogenic niches. L-T_4_ rescued the number of DCX+ cells while L-T_4_+T_1_AM caused a 30.61% rebound relative to the euthyroid state. Pair-wise comparisons between the five groups using a *t*-test and Bonferroni-corrected for multiple comparisons (** *p* < 0.01, n.s. not significant). Hypothyroid *n* = 4, Euthyroid *n* = 4, L-T_4_
*n* = 4, L-T_4_+T_1_AM *n* = 4, T_1_AM *n* = 4.

**Figure 5 ijms-24-13845-f005:**
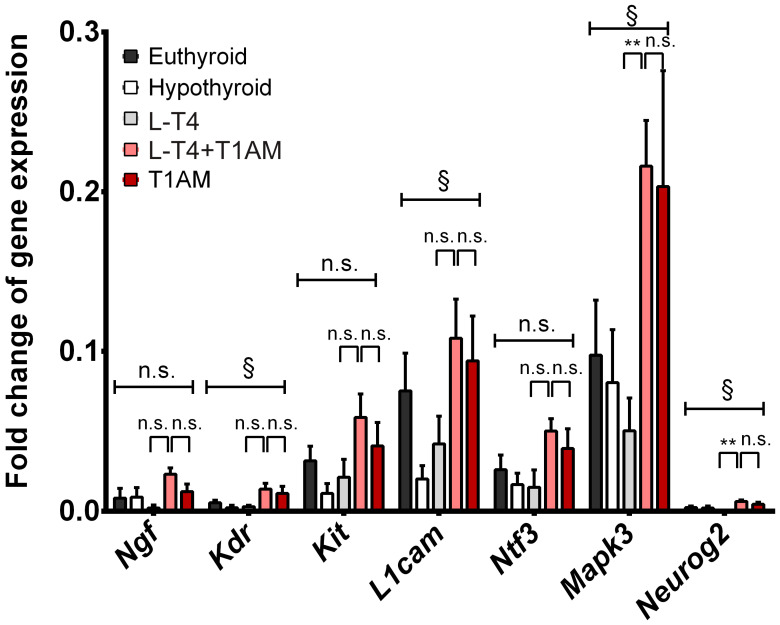
Analysis of expression of genes involved in neurogenetic pathways. One-way ANOVA showed significant between-treatment differences in gene expression (§ *p* < 0.05). *t*-test pair-wise comparisons were performed to test the differences between L-T_4_+T_1_AM and either L-T_4_ or T_1_AM (** *p* < 0.01, n.s. not significant). Hypothyroid *n* = 5, euthyroid *n* = 4, L-T_4_
*n* = 4, L-T_4_+T_1_AM *n* = 7, T_1_AM *n* = 6.

**Table 1 ijms-24-13845-t001:** Measures of anxiety-related behaviours, spontaneous locomotor activity, memory, and depression-related behaviours in the mouse model of hypothyroidism and replacement treatments.

Experiment	Hypothyroid	Euthyroid	L-T_4_	L-T_4_+T_1_AM	T_1_AM	F (DFn, DFd), *p*
**EPM:**						
% of time spent in open arms	7.51 ± 0.99	9.81 ± 2.85	11.83 ± 2.65	9.34 ± 1.67	7.06 ± 1.77	0.79 (4, 57), 0.79
*n* of entries into open arms	3.17 ± 0.49	Mdn = 3 IQR = 2–4	4.08 ± 0.73	2.92 ± 0.43	2.33 ± 0.51	3.69, 0.45 ^a^
*n* of entries into closed arms	7.00 ± 0.73	9.67 ± 0.64	8.83 ± 0.76	7.69 ± 0.87	8.58 ± 0.72	1.87 (4, 56), 0.13
*n* of total head dips	11.67 ± 2.55	Mdn = 15 IQR = 10–20	13.25 ± 2.23	11.23 ± 2.09	12.25 ± 2.10	2.39, 0.66 ^a^
*n* of unprotected head dips	4.30 ± 0.89	5.38 ± 2.40	5.50 ± 1.39	4.10 ± 1.00	3.50 ± 0.98	0.29 (4, 48), 0.89
*n* of protected head dips	9.70 ± 1.71	11.23 ± 1.21	10.40 ± 1.17	10.50 ± 1.21	11.2 ± 0.87	0.25 (4, 48), 0.25
*n* of end arm explorations	1.10 ± 0.35	1.31 ± 0.41	1.10 ± 0.50	0.45 ± 0.21	0.90 ± 0.35	0.77 (4, 49), 0.55
*n* of stretch attend postures	2.60 ± 0.65	6.69 ± 1.07	6.40 ± 1.51	6.50 ± 1.81	4.60 ± 0.87	1.96 (4, 48), 0.12
**OF:**						
total distance travelled (m)	789.1 ± 79.61	996.8 ± 65.35	846.7 ± 90.59	943.6 ± 84.08	Mdn = 901.9 IQR = 694.8–1025	3.77, 0.44 ^a^
time spent in open vs. closed arms (s)	Mdn = 0.81 IQR = 0.70–0.86	0.79 ± 0.02	0.85 ± 0.02	0.79 ± 0.04	0.83 ± 0.04	2.97, 0.56 ^a^
**ORT:**						
discrimination index	0.02 ± 0.09	0.29 ± 0.06	0.27 ± 0.08	0.34 ± 0.08	−0.01 ± 0.10	3.15 (3, 42), 0.03
**TST:**						
% of immobility time	61.86 ± 1.73	59.03 ± 2.52	62.13 ± 2.83	62.1 ± 2.88	60.72 ± 1.84	0.32 (4, 43), 0.86

^a^ Differences between groups were assessed with the non-parametric Kruskal–Wallis test.

**Table 2 ijms-24-13845-t002:** Behavioural interpretation of the variables recorded in the elevated plus maze test.

Variable	Behavioural Interpretation
Time in open arms (%)	Anxiety-related
*n* of entries into open arms	Anxiety-related
*n* of entries into closed arms	Spontaneous locomotor activity
Total *n* of head dips	Anxiety-related
*n* of unprotected head dips	Anxiety-related
*n* of protected head dips	Decision-making; assessment of height and openness
*n* of end-arm explorations	Anxiety-related
*n* of SAPs	Risk-assessment

## Data Availability

The datasets generated and analysed during the current study are available from the corresponding author on reasonable request.

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
