# Peer review of "Effect of Combined Levothyroxine (L-T4) and 3-Iodothyronamine (T1AM) Supplementation on Memory and Adult Hippocampal Neurogenesis in a Mouse Model of Hypothyroidism"

_ijms, 2023, doi:10.3390/ijms241813845_

Round 1

Reviewer 1 Report

The authors used a mouse model of thyroid hormone deficiency and investigated the effect of supplementation with either T4 or T4 plus the metabolite T1AM on behaviour on the plus maze, in the open field, in a novel object recognition test protocol, and on neurogenesis and gene expression markers. The paper is generally well written although there are a few English language errors (see below).  

First of all it need to be mentioned that most of the lettering and symbols in the figures is far too small and near-impossible to read, particularly after printing on A4. The authors need to correct their figures such, that when reduced in size for printing, the symbols and letters are still clear.

With respect to the manuscript itself, my main concern here is the conclusion of the paper that T1AM effects enhance those of T4 and do so in a “synergistic” manner. The only result where this may be correct is the gene expression analysis where T4 on its own had little effect but addition of T1AM markedly increased the expression of some of the genes studied. Unfortunately, that part of the analysis did not include a crucial T1AM-only group (it is nowhere explained why) so it remains possible that the changes in gene expression can be induced by T1AM on its own, i.e. not a synergistic effect of the combination with T4. Elsewhere, there is no evidence whatsoever of a synergistic effect of T4 and T1AM. In figure 2E, the effect of T4 is the same as that of T4 + T1AM, in other words T1AM had no effect at all. The same is seen in figure 1C and 3B. In figure 3B the number of DCX-positive cells was a little bit higher after T4 + T1AM treatment than after T4-only treatment, but this was not a significant difference. The authors will have to remove any mention of T1AM having synergistic effects or explain why they come to this conclusion.

A second concern is the conclusion that the T4 and T1AM combination has its effect “through an effect on hippocampal neurogenesis” (for example line 146 and 169). This causality is actually not shown. There may well be an effect of the T4/T1AM treatment on behaviour and on hippocampal neurogenesis but it remains to be seen whether the latter causes the former. Only additional studies where somehow neurogenesis is blocked during T4/T1AM treatment would be able to show that the treatment needs the changes in neurogenesis to exert its action on behaviour. The effects on behaviour and on neurogenesis could be entirely independent. At minimum, this needs to be added to the discussion as a limitation and topic of future studies.

The legend to figure 1 first mentions group sizes of 15 or greater, yet for the behavioural analysis at the end of this legend the n was only 6-9. It is not clear why there was such a difference.  

In line 124, 126, 177 and elsewhere in the manuscript, findings in the novel object recognition test are interpreted as “hippocampus-dependent memory”. Depending on how the ORT is done, the hippocampus may be involved, but so are other brain regions. I strongly recommend removing this interpretation from the Results section and only mentioning this as one possible interpretation of the ORT data in the Discussion. It would also be good to see why the authors chose the ORT and not other, more spatial tasks which are more traditionally linked to hippocampal function. In this regard it is also important to note that the authors used only one minute as the interval between the Familiarization and Test stages of the test. This is very short and it is possible that the ORT result is more dependent on novelty than on long-term memory.

Other and specific corrections:

Line 23, 40, 63 and elsewhere: correct “performances” to “performance”

Line 118: delete “probably due to lack of statistical power”. Comments like this are for the Discussion section.

Line 63: remove “even more if compared to L-T4 hypothyroid treated mice”. This is a subjective interpretation of the data and not based on statistical differences between the groups.

Line 171: correct “anecdotical”

Line 172-175: the discussion here is unclear. If there was residual thyroperoxidase activity and normal T3 levels, why does that explain the lack of changes in anxiety and depression-like behaviour but is not relevant for the ORT results?

 Line 176: here the authors say that their model is one of “moderate” hypothyroidism. What is that based upon? T4 levels were almost entirely suppressed.

Line 182-183: this sentence can be read as stating that the T4 treatment was sufficient to reverse the behavioural effects of hypothyroidism but the T1AM additional treatment had no effect. Yet the authors in line 192 keep saying that T1AM “further” improved the effect of T4. This is not actually seen in the data and should be deleted here. In figure 2E there is no difference between T4 and T4 + T1AM. Similarly, in figure 3B there is no significant difference between the effect of the two treatments. As mentioned earlier, the term “synergistic” (line 194) is not correct here.

Line 222-232 discuss the low number of animals per group although it does not really become clear why this is the case. Did mice die because of the chronic treatments?  Using the 3R principles here seems unconvincing. If a study has low n per group and no clear conclusions can be drawn from the results, more animals are used for nothing than if the group sizes are larger and a clear result is found.

Line 233: again, the authors mention “moderate” TH changes but it is unclear what this is based upon. It is also never mentioned in the aims of the study.

Line 236: as mentioned above, comments like “gave a further boost” are not supported by the data and should be removed. Similar for “synergistic interaction” which is not supported by the data.

Section 4.1.: what is “die”? I presume the authors mean “day”? The authors have to add references why they chose these doses of the treatments.

Section 4.3.: I recommend adding a time line diagram of when the behavioural tests were done and how much time there was between them.

Line 260: how were the mice “habituated” to the room?

Section 4.3: include the size of the arms and the wall of the EPM. Include light level at the surface of the maze and the OF.

Line 283: suspended from a shelf.

See my report.

Author Response

We would like to thank the Editor and Reviewers for their constructive criticism and suggestions.

We have revised our manuscript, according to their indications, as detailed below. Please note that

the changes made in the manuscript are shown in blue.

# Reviewer 1

The authors used a mouse model of thyroid hormone deficiency and investigated the effect of supplementation with either T4 or T4 plus the metabolite T1AM on behaviour on the plus maze, in the open field, in a novel object recognition test protocol, and on neurogenesis and gene expression markers. The paper is generally well written although there are a few English language errors (see below).  

We thank the Reviewer for their positive feedback.

  1. First of all it need to be mentioned that most of the lettering and symbols in the figures is far too small and near-impossible to read, particularly after printing on A4. The authors need to correct their figures such, that when reduced in size for printing, the symbols and letters are still clear.

We apologise for the poor quality of the figures. We have increased the font size in all figures. To further improve clarity, we have split the results of behavioural tests (previously shown in Figure 2) across two separate figures:

  • Figure 2. Effect of hypothyroidism and different replacement treatments on locomotion.
  • Figure 3. Effect of hypothyroidism and different replacement treatments on memory.

We have also put the results of gene expression analysis into a separate figure:

  • Figure 5. Analysis of expression of genes involved in neurogenetic pathways.

  1. With respect to the manuscript itself, my main concern here is the conclusion of the paper that T1AM effects enhance those of T4 and do so in a “synergistic” manner. The only result where this may be correct is the gene expression analysis where T4 on its own had little effect but addition of T1AM markedly increased the expression of some of the genes studied. Unfortunately, that part of the analysis did not include a crucial T1AM-only group (it is nowhere explained why) so it remains possible that the changes in gene expression can be induced by T1AM on its own, i.e. not a synergistic effect of the combination with T4. Elsewhere, there is no evidence whatsoever of a synergistic effect of T4 and T1AM. In figure 2E, the effect of T4 is the same as that of T4 + T1AM, in other words T1AM had no effect at all. The same is seen in figure 1C and 3B. In figure 3B the number of DCX-positive cells was a little bit higher after T4 + T1AM treatment than after T4-only treatment, but this was not a significant difference. The authors will have to remove any mention of T1AM having synergistic effects or explain why they come to this conclusion.

We would like to thank the Reviewer for their insightful comment. We agree that our behavioural and immunofluorescence data do not fully support the conclusion that T1AM effects enhance those of L-T4 in a synergistic manner. We have now removed every mention of a synergistic interaction between L-T4 and T1AM, as detailed below:

  • The title was edited to: “Effect of combined levothyroxine (L-T4) and 3-iodothyronamine (T1AM) supplementation on memory and adult hippocampal neurogenesis in a mouse model of hypothyroidism”.
  • The abstract conclusion was edited to: “Our data confirm that L-T4 is necessary and sufficient to recover memory and hippocampal neurogenesis deficits associated with hypothyroidism, while we found no evidence to support the role of non-canonical TH signalling.”.
  • In paragraph 2.4., the title was edited to “L-T4 and L-T4 + T1AM restored the number of DCX+ cells” and the mention of a synergistic effect was replaced with “These results confirm that L-T4 is a crucial endocrine signal controlling progenitor development within neurogenic niches, while there are no significant effects of T1AM in the modulation of adult hippocampal neurogenesis.”.
  • In the discussion, the initial sentence summarizing our findings was edited to: “In this study, we confirmed that L-T4 monotherapy is necessary and sufficient to recover deficits in memory and adult hippocampal neurogenesis associated with adult-onset hypothyroidism. We observed no further improvement in neurocognitive and neurobiological alterations from augmenting L-T4 with T1AM.”. We removed mention of a synergistic interaction between canonical and non-canonical TH pathways from lines 217-220.

In addition, following the Reviewer’s suggestion, we have now included the T1AM-only group in our gene expression analysis. We reported the new findings in paragraph 2.5., as follows:

“The qPCR analysis revealed differentially expressed genes in neurogenetic pathways. One-way ANOVA revealed a significant effect of treatment on expression of Kdr (F (4,24) = 3.20, p = 0.03), L1cam (F (4,24) = 2.84, p = 0.04), Mapk3 (F (4,23) = 2.96, p = 0.04) and Neurog2 (F (4,20) = 3.94, p = 0.02). We observed a trend towards an effect of treatment on expression of Ngf (F (4,21) = 2.50, p = 0.07), Kit (F (4,24) = 2.51, p = 0.07) and Ntf3 (F (4,23) = 2.51, p = 0.07).”

To test the hypothesis that changes in gene expression were induced by L-T4 and T1AM synergistic effect versus T1AM on its own, we used pair-wise t-tests to compare: (1) L-T4 and L-T4+T1AM; (2) T1AM and L-T4+T1AM. After Bonferroni-correcting for multiple comparisons, we found significantly upregulated Mapk3 and Neurog2 in L-T4+T1AM-treated mice as compared to L-T4, while we could not detect any significant differences between L-T4+T1AM and T1AM, thus suggesting that changes in gene expression may be induced by T1AM on its own. These findings were reported in paragraph 2.5., as follows:

“We performed t-test pair-wise comparisons to test the differences between L-T4+T1AM and either L-T4 or T1AM, using Bonferroni correction for multiple comparisons (0.05/14=0.004). We found that Mapk3 and Neurog2 were significantly upregulated in L-T4 + T1AM-treated relative to L-T4-treated mice (Mapk3, t = 4.51, pcorr = 0.002; Neurog2, t = 4.81, pcorr = 0.001), with non-significant differences between L-T4+T1AM and T1AM (Mapk3, t = 0.16, puncorr = 0.87; Neurog2, t = 1.13, puncorr = 0.29), suggesting that changes in gene expression could be induced by T1AM on its own.”.

The corresponding figure (Figure 5 in the revised manuscript) was updated accordingly.

Finally, we have reworded our conclusion to provide a more balanced interpretation of our findings, as follows:

“Our data show that moderate TH changes are sufficient to induce significant impairments in hippocampal related memory and neurogenesis. L-T4 was necessary and sufficient to recover both memory and number of neuroprogenitors to the euthyroid state, while we found no evidence to support the role of non-canonical TH signalling, despite an effect of T1AM on expression of genes involved in neurogenetic pathways, that should be further explored in future studies.”.

  1. A second concern is the conclusion that the T4 and T1AM combination has its effect “through an effect on hippocampal neurogenesis” (for example line 146 and 169). This causality is actually not shown. There may well be an effect of the T4/T1AM treatment on behaviour and on hippocampal neurogenesis but it remains to be seen whether the latter causes the former. Only additional studies where somehow neurogenesis is blocked during T4/T1AM treatment would be able to show that the treatment needs the changes in neurogenesis to exert its action on behaviour. The effects on behaviour and on neurogenesis could be entirely independent. At minimum, this needs to be added to the discussion as a limitation and topic of future studies.

We agree with the Reviewer that more experiments would be needed to show the causal relationship between effects on behaviour and on hippocampal neurogenesis. We have now presented the effects on memory and neurogenesis as independent, and have included this to the limitations of the present study, as follows:

  • Lines 259-262: “First, our findings do not clarify if the effects of replacement treatments on memory are causally mediated by changes in hippocampal neurogenesis. Future studies should investigate if blocking neurogenesis during L-T4 and/or L-T4+T1AM treatment prevents memory rescue.”.

  1. The legend to figure 1 first mentions group sizes of 15 or greater, yet for the behavioural analysis at the end of this legend the n was only 6-9. It is not clear why there was such a difference.  

Thank you for spotting this. Figure 1 presents the experimental protocol, body weight and hormonal determinations, while it does not contain behavioural analyses. The sample sizes in panel 1 (euthyroid, n=17; hypothyroid, n=15; L-T4, n=18; L-T4 + T1AM, n=17; and T1AM, n=15) correspond to the total number of mice entering the protocol of hypothyroidism induction and replacement treatment. Panels C and D report the determinations of serum total T4 and total T3 done with the HPLC-MS/MS method, reported in paragraph 4.2. and described in detail in the supplementary material. We also determined serum free T4 and free T3 using an AIA 2000LA immunoassay platform, used for human samples in the laboratories of the CNR Institute of Clinical Physiology. However, as this platform is set up for human samples, we had to pool 2 to 3 samples to obtain a sufficient volume for analysis. Moreover, this did not always work, so we ended up with very small sample sizes for serum free T4 and free T3 and decided to not present these analyses in the final manuscript. These samples could not be used for analysis with the HPLC-MS/MS method too. This explains the different sample size between mice entering the protocol and number of samples used for hormonal determination (panels C and D). We have now clarified this in legend to figure 1, as follows:

“The remaining samples were used to analyse serum fT4 and fT3 on an AIA 2000LA immunoassay platform (data not shown).”

  1. In line 124, 126, 177 and elsewhere in the manuscript, findings in the novel object recognition test are interpreted as “hippocampus-dependent memory”. Depending on how the ORT is done, the hippocampus may be involved, but so are other brain regions. I strongly recommend removing this interpretation from the Results section and only mentioning this as one possible interpretation of the ORT data in the Discussion. It would also be good to see why the authors chose the ORT and not other, more spatial tasks which are more traditionally linked to hippocampal function. In this regard it is also important to note that the authors used only one minute as the interval between the Familiarization and Test stages of the test. This is very short and it is possible that the ORT result is more dependent on novelty than on long-term memory.

We agree with the referee that the ORT involves different areas for the formation of the memory of a familiar object and we removed the interpretation of “hippocampal dependent memory”. We used the less cognitively demanding task, without introducing object/place or object/place/context associations, as it was demonstrated that lesion of the entorhinal cortex would affect only the associative memory but not the recognition memory (Wilson et al., 2013), which is preserved by intact hippocampus. We chose the ORT because we have previously found a good correlation between the onset of synaptic deficit in hippocampal circuit and the development of object recognition memory impairment in a mouse model of neurodegeneration (Criscuolo et al., 2017).

Other and specific corrections:

  1. Line 23, 40, 63 and elsewhere: correct “performances” to “performance”

Thank you. We have corrected this throughout the manuscript.

  1. Line 118: delete “probably due to lack of statistical power”. Comments like this are for the Discussion section.

We deleted “probably due to lack of statistical power” from the result session. This issue was already discussed in the limitation session within the discussion.

  1. Line 63: remove “even more if compared to L-T4 hypothyroid treated mice”. This is a subjective interpretation of the data and not based on statistical differences between the groups.

We deleted this sentence in the new presentation of gene expression analysis. See point 2 for further details.

  1. Line 171: correct “anecdotical”

We replaced “anecdotical” with “empirical”.

  1. Line 172-175: the discussion here is unclear. If there was residual thyroperoxidase activity and normal T3 levels, why does that explain the lack of changes in anxiety and depression-like behaviour but is not relevant for the ORT results?

This is a good point, for which we do not have a definite answer. Based on clinical observations and previous reports, we expected to observe alterations in measures of depression and anxiety in our model of hypothyroidism. We explained the lack of effects on these measures with the limited impact of our hypothyroidism induction protocol on serum TT3 levels, which remained virtually unchanged. To explain that memory deficits could be observed following the same hypothyroidism induction, we speculated that “cognitive functions are more susceptible to TH deficiency” (lines 200-201).

  1. Line 176: here the authors say that their model is one of “moderate” hypothyroidism. What is that based upon? T4 levels were almost entirely suppressed.

We agree with the Reviewer that our use of “moderate” was confusing, therefore we introduced the adjective in the preceding sentence (line 196), as follows:

“Our hypothyroidism induction protocol almost completely suppressed serum TT4 levels, while we observed no changes in serum TT3 levels, suggesting that our protocol induced moderate hypothyroidism.”

  1. Line 182-183: this sentence can be read as stating that the T4 treatment was sufficient to reverse the behavioural effects of hypothyroidism but the T1AM additional treatment had no effect. Yet the authors in line 192 keep saying that T1AM “further” improved the effect of T4. This is not actually seen in the data and should be deleted here. In figure 2E there is no difference between T4 and T4 + T1AM. Similarly, in figure 3B there is no significant difference between the effect of the two treatments. As mentioned earlier, the term “synergistic” (line 194) is not correct here.

We removed every mention of a synergistic effect and similar wording, as detailed in point 2. We believe our manuscript now provides a more balanced interpretation of our findings.

  1. Line 222-232 discuss the low number of animals per group although it does not really become clear why this is the case. Did mice die because of the chronic treatments?  Using the 3R principles here seems unconvincing. If a study has low n per group and no clear conclusions can be drawn from the results, more animals are used for nothing than if the group sizes are larger and a clear result is found.

The Reviewer’s comment made us realise that the text of the limitation session was not clear. To improve clarity, we added some adverbs: first, second, and finally. The 3R principle was not used to justify the small sample size. We provided justification for our sample size using power calculations. However, our statistical analysis was powered to detect larger effect sizes, based on previous reports in thyroidectomized or transgenic rodents, while we observed changes of small to moderate effect size, probably because our protocol did not suppress TT3 levels (moderate hypothyroidism). This is explained in lines 263-267. The 3R principle was instead used to support aborting the objective to measure endogenous T1AM in brain tissues. We could not detect a clear T1AM endogenous signal in most of our pilot brain samples, meaning that T1AM levels were below the detection limit for most of the samples. Therefore, we decided to abort this objective, that would have required at least 30-40 more mice with a high risk of not getting usable T1AM measures. We deleted reference to the 3R principle and clarified that we did not pursue the objective for technical reasons.  

  1. Line 233: again, the authors mention “moderate” TH changes but it is unclear what this is based upon. It is also never mentioned in the aims of the study.

Please refer to point 11.

  1. Line 236: as mentioned above, comments like “gave a further boost” are not supported by the data and should be removed. Similar for “synergistic interaction” which is not supported by the data.

Please refer to point 2.

  1. Section 4.1.: what is “die”? I presume the authors mean “day”? The authors have to add references why they chose these doses of the treatments.

We have replaced die (Latin for day) with day for clarity.

The dose of methimazole and potassium perchlorate used in this study was chosen based on: Hackenmueller SA, Marchini M, Saba A, Zucchi R, Scanlan TS. Biosynthesis of 3-iodothyronamine (T1AM) is dependent on the sodium-iodide symporter and thyroperoxidase but does not involve extrathyroidal metabolism of T4. Endocrinology. 2012 Nov;153(11):5659-67. The final dose of 0.20 mg/g/day for methimazole and 0.30 mg/g/day for potassium perchlorate were estimated considering the respective initial concentrations of 0.1% and 0.2% in drinking water, average daily water intake of 5 ml and average body weight of 30 g.

The dose of L-T4 was chosen based on: Panici JA, Harper JM, Miller RA, Bartke A, Spong A, Masternak MM. Early life growth hormone treatment shortens longevity and decreases cellular stress resistance in long-lived mutant mice. FASEB J 2010; 24(12): 5073-5079. In this report a dose of 0.1 μg/g BW 3x/week was used. Therefore we used 0.1 μg/g BW/(7 week days/3) = 0.1 μg/g BW/2.3 days = 0.04 μg/g BW/day.

The dose of T1AM was calculated based on: Bellusci L, Laurino A, Sabatini M, Sestito S, Lenzi P, Raimondi L, Rapposelli S, Biagioni F, Fornai F, Salvetti, A, et al. New Insights into the Potential Roles of 3-Iodothyronamine (T1AM) and Newly Developed Thyronamine-Like TAAR1 Agonists in Neuroprotection. Front. Pharmacol. 2017, 8, 905, doi:10.3389/fphar.2017.00905. In absence of previous data on chronic T1AM administration, the dose for the present study was calculated multiplying by 28 (length of replacement treatment, in days) the minimum acute dose of T1AM which was reported to improve cognition in mice, 4 μg/kg BW.

  1. Section 4.3.: I recommend adding a time line diagram of when the behavioural tests were done and how much time there was between them.

We have specified the order and timing of behavioural tests in paragraph 4.3, as follows:

“Afterwards, mice underwent behavioural tests over three consecutive days, in the follow-ing order: day 1, EPM; day 2, OF and ORT; day 3, TST.”

  1. Line 260: how were the mice “habituated” to the room?

The experimenter moved them from their usual holding room to the testing room. We have clarified this is the manuscript.

  1. Section 4.3: include the size of the arms and the wall of the EPM. Include light level at the surface of the maze and the OF.

These details were added to section 4.3. The elevated plus maze (EPM) apparatus consists of 4 arms (25 cm length x 5 cm width) forming a plus sign, elevated 50 cm above the floor. Two opposite arms have walls (16 cm height, closed arms), while the other two do not (open arms). The testing room is a soundproof environment with 100 lux illumination level.

  1. Line 283: suspended from a shelf.

Thank you. We have corrected this.

Reviewer 2 Report

Review on the manuscript of Rutigliano G et al.: “Combined levothyroxine (L-T4) and 3-iodothyronamine (T1AM) supplementation improves memory and adult hippocampal neurogenesis in a mouse model of hypothyroidism”.

In this manuscript, authors explored, in a pharmacological hypothyroid mouse model, whether the administration of Levothyroxine (L-T4) with T1AM would improve behavioral correlates of depression, anxiety, and memory through an effect on hippocampal neurogenesis. The manuscript reports that hypothyroid mice show impaired performances in the recognition of novel objects and reduced number of neuroprogenitors in hippocampal neurogenic niches. These effects were recovered after L-T4+T1AM administration. In addition, qPCR analysis revealed an upregulation of Ngf, Kdr, Ntf3, Mapk3 and Neu-rog2 genes in L-T4+T1AM treated mice, suggesting that these genes could be involved in the rescue phenotype.

The manuscript is very clear and well written. In addition, I consider that the manuscript is precise on the questions that authors proposed to answer. Thus, the issues that arise to me are listed below, so, I hope the authors find the following comments and suggestions useful.

1 – I recommend authors to increase the font size in figures. In addittion, the colors in the graph 1B are difficult to follow. Can authors use different colors for the different conditions to make the graph easier to understand.

2 – I recommend authors to cite the data of Figure S2 in the text (in the current version of the manuscript, these data are not described in the text).

3 – From the data shown in the manuscript, authors concluded that L-T4 was necessary and sufficient to recover both memory and number of neuroprogenitors to the euthyroid state, but the combination with T1AM gave a further boost, suggesting a synergistic interaction between canonical and non-canonical TH signalling. Do authors have any suggestion on the non-canonical TH signalling pathways that could be implicated in such phenotype. It would be good to discuss this topic in more detail.

4 – What are the authors’ expectations for the effect of L-T4+T1AM administration in a severe hypothyroidism? Would it be the same? Can authors complement the discussion with some expectations?

5 - To test the hypothesis that some of hypothyroidism-induced cognitive impairments could be explained by T1AM deficiency, authors attempted to measure brain T1AM, but found no clear endogenous signal (below detection limit). Would it be possible to do it in animals with hyperthyroidism? If it is possible, it would give another level of knowledge.

Author Response

We would like to thank the Editor and Reviewers for their constructive criticism and suggestions.

We have revised our manuscript, according to their indications, as detailed below. Please note that the changes made in the manuscript are shown in blue.

Review on the manuscript of Rutigliano G et al.: “Combined levothyroxine (L-T4) and 3-iodothyronamine (T1AM) supplementation improves memory and adult hippocampal neurogenesis in a mouse model of hypothyroidism”.

In this manuscript, authors explored, in a pharmacological hypothyroid mouse model, whether the administration of Levothyroxine (L-T4) with T1AM would improve behavioral correlates of depression, anxiety, and memory through an effect on hippocampal neurogenesis. The manuscript reports that hypothyroid mice show impaired performances in the recognition of novel objects and reduced number of neuroprogenitors in hippocampal neurogenic niches. These effects were recovered after L-T4+T1AM administration. In addition, qPCR analysis revealed an upregulation of Ngf, Kdr, Ntf3, Mapk3 and Neu-rog2 genes in L-T4+T1AM treated mice, suggesting that these genes could be involved in the rescue phenotype.

The manuscript is very clear and well written. In addition, I consider that the manuscript is precise on the questions that authors proposed to answer. Thus, the issues that arise to me are listed below, so, I hope the authors find the following comments and suggestions useful.

We thank the Reviewer for their positive feedback.

  1. I recommend authors to increase the font size in figures. In addittion, the colors in the graph 1B are difficult to follow. Can authors use different colors for the different conditions to make the graph easier to understand.

We apologise for the poor quality of the figures. We have increased the font size and changed colours in all figures. To further improve clarity, we have split the results of behavioural tests (previously shown in Figure 2) across two separate figures:

  • Figure 2. Effect of hypothyroidism and different replacement treatments on locomotion.
  • Figure 3. Effect of hypothyroidism and different replacement treatments on memory.

We have also put the results of gene expression analysis into a separate figure:

  • Figure 5. Analysis of expression of genes involved in neurogenetic pathways.

  1. I recommend authors to cite the data of Figure S2 in the text (in the current version of the manuscript, these data are not described in the text).

We thank the Reviewer for highlighting this. We have now clarified in lines 115-116 that locomotor activity and thigmotaxis were measured in the open field test while % immobility time (plotted in Figure S2.B) was measured in the tail suspension test, as follows:

“We did not find any significant differences in spontaneous locomotor activity and thigmotaxis in the open field test (OF), and % immobility time in the tail suspension test (TST) between treatments.”

TST data are also reported in table 1 (mean ± SEM).

  1. From the data shown in the manuscript, authors concluded that L-T4 was necessary and sufficient to recover both memory and number of neuroprogenitors to the euthyroid state, but the combination with T1AM gave a further boost, suggesting a synergistic interaction between canonical and non-canonical TH signalling. Do authors have any suggestion on the non-canonical TH signalling pathways that could be implicated in such phenotype. It would be good to discuss this topic in more detail.

Following Reviewer 1’s recommendation, we revised our conclusions to provide a more balanced interpretation of our findings. In fact, our behavioural and immunofluorescence data do not fully support the conclusion that T1AM effects enhance those of L-T4 in a synergistic manner. Nonetheless, we found that T1AM may induce changes in gene expression on its own. We have previously demonstrated that T1AM has neuroprotective actions in models of amyloidosis, ischemia-reperfusion injury and neuroinflammation, and that these are replicated by TAAR1 (trace amine-associated receptor 1) agonists and abolished by TAAR1 antagonists. Therefore, we hypothesize that T1AM actions are mediated by TAAR1. Although testing the mediating role of TAAR1 in T1AM effects in our hypothyroidism model was beyond the scope of this study, we offered a brief overview of the signalling pathway activated by TH derivatives in the discussion, as suggested by Reviewer 2, as follows:

“T1AM does not bind TR but to several molecular targets, including the trace amine-associated receptor 1 (TAAR1), alpha2 adrenergic receptors, transient receptor potential channels, and ApoB-100 [51]. How its distinct target profile links to gene expression will need to be clarified in future studies. However, we hypothesise a role for TAAR1, due to growing evidence of its involvement in neuropsychiatric disorders [52]. Furthermore, we have previously demonstrated that T1AM neuroprotective actions in models of amyloidosis, ischemia-reperfusion injury, and neuroinflammation are dependent on TAAR1. TAAR1 was reported to activate distinct signalling pathways depending on its localization in different cellular compartments. When on the plasma membrane, TAAR1 activation leads to increased intracellular cAMP levels and activation of protein kinase A, through coupling to Gαs proteins and adenylyl cyclase activation. Intracellular TAAR1 activation induces its coupling with Gα13 subunits to activate the GTPase RhoA. TAAR1 activation is also associated with G protein-coupled inwardly rectifying potassium chan-nels and G protein-independent inhibition of glycogen synthase kinase 3β.”

We have added the following supporting references:

Borowsky B. et al.Trace amines: identification of a family of mammalian G protein-coupled receptors. Proc. Natl. Acad. Sci. U. S. A. 2001; 98: 8966-8971

Lindemann L. et al. Trace amine-associated receptors form structurally and functionally distinct subfamilies of novel G protein-coupled receptors. Genomics. 2005; 85: 372-385

Grandy D.K. Trace amine-associated receptor 1-family archetype or iconoclast? Pharmacol. Ther. 2007; 116: 355-390

Underhill S.M. et al. Amphetamines signal through intracellular TAAR1 receptors coupled to Gα13 and GαS in discrete subcellular domains. Mol. Psychiatry. 2021; 26: 1208-1223

Bradaia A. et al. The selective antagonist EPPTB reveals TAAR1-mediated regulatory mechanisms in dopaminergic neurons of the mesolimbic system. Proc. Natl. Acad. Sci. U. S. A. 2009; 106: 20081-20086

Yang W. et al. Dopamine evokes a trace amine receptor-dependent inward current that is regulated by AMP kinase in substantia nigra dopamine neurons. Neuroscience. 2020; 427: 77-91

Harmeier A. et al. Trace amine-associated receptor 1 activation silences GSK3β signaling of TAAR1 and D2R heteromers. Eur. Neuropsychopharmacol. 2015; 25: 2049-2061

  1. What are the authors’ expectations for the effect of L-T4+T1AM administration in a severe hypothyroidism? Would it be the same? Can authors complement the discussion with some expectations?

This is an interesting question. However, at this stage, we do not have enough elements to support a hypothesis. We therefore prefer to not include comments on the effects of combined L-T4+T1AM administration in a model of severe hypothyroidism, as any comments would be highly speculative. We plan to conduct more experiments to investigate this issue in the future.

  1. To test the hypothesis that some of hypothyroidism-induced cognitive impairments could be explained by T1AM deficiency, authors attempted to measure brain T1AM, but found no clear endogenous signal (below detection limit). Would it be possible to do it in animals with hyperthyroidism? If it is possible, it would give another level of knowledge.

Thank you for your suggestion. The biosynthetic pathway leading to T1AM in the brain remains mysterious. It has been demonstrated that endogenous liver T1AM concentration is significantly reduced in hypothyroid mice, and remains low even following adequate TH supplementation, suggesting that T1AM biosynthesis requires the integrity of the thyroid gland. On the other hand, extrathyroidal T1AM synthesis from T4 was demonstrated in a rat intestine preparation. T1AM formation from T4 relied on deiodinases (Dio) and ornithine decarboxylase (ODC). OCD was shown to decarboxylate T4 and 3,5-T2 to the corresponding thyronamines T4AM and 3,5-T2AM. If this biosynthesis pathways were confirmed in the brain, we would expect to detect higher T1AM concentrations in the brain of hyperthyroid mice. However, it is also possible that other regulatory factors contribute to determine the rate of T1AM production from T4 in the brain. For example, we observed increased release of T1AM and its metabolite 3-iodothyroacetic acid (TA1) during oxygen-glucose deprivation. To further complicate this matter, T1AM biosynthetic pathway would require the orchestrated crosstalk of astrocyte and neurons. We are actively investigating this topic using pharmacological manipulations of putative enzymes in organotypic brain slices and hope to have some interesting results soon. In brief, it cannot be assumed that T1AM levels would be higher in the hyperthyroid condition, due to the intricacies of TH metabolic pathways.

Round 2

Reviewer 1 Report

The authors have made many changes to their manuscript which has improved considerably. The conclusion of the paper is now essentially the opposite of what it was in the first version and the authors should be commended on accepting all my comments and changing their manuscript accordingly. There is just one instance where the authors still mention "hippocampus-dependent" (line 216-217. I suggest to remove this. Other than that, I have no further questions or comments.

Author Response

We have removed the outstanding instance of "hippocampus-dependent", thank you. 

We would like to sincerely thank the Reviewer for the attention they have dedicated to review our manuscript.